# Cone-constrained Principal Component Analysis

**Yash Deshpande**
Electrical Engineering
Stanford University

**Andrea Montanari**
Electrical Engineering and Statistics
Stanford University

**Emile Richard**
Electrical Engineering
Stanford University

## Abstract

Estimating a vector from noisy quadratic observations is a task that arises naturally in many contexts, from dimensionality reduction, to synchronization and phase retrieval problems. It is often the case that additional information is available about the unknown vector (for instance, sparsity, sign or magnitude of its entries). Many authors propose non-convex quadratic optimization problems that aim at exploiting optimally this information. However, solving these problems is typically NP-hard.

We consider a simple model for noisy quadratic observation of an unknown vector $\mathbf{v_0}$. The unknown vector is constrained to belong to a cone $\mathcal{C} \ni \mathbf{v_0}$. While optimal estimation appears to be intractable for the general problems in this class, we provide evidence that it is tractable when $\mathcal{C}$ is a convex cone with an efficient projection. This is surprising, since the corresponding optimization problem is non-convex and –from a worst case perspective– often NP hard. We characterize the resulting minimax risk in terms of the statistical dimension of the cone $\delta(\mathcal{C})$. This quantity is already known to control the risk of estimation from gaussian observations and random linear measurements. It is rather surprising that the same quantity plays a role in the estimation risk from quadratic measurements.

## 1 Introduction

In many statistical estimation problems, observations can be modeled as noisy quadratic functions of an unknown vector $\mathbf{v_0} = (\mathbf{v}_{0,1}, \mathbf{v}_{0,2}, \ldots, \mathbf{v}_{0,n})^\mathsf{T} \in \mathbb{R}^n$. For instance, in positioning and graph localization [5, 24], one is given noisy measurements of pairwise distances $(\mathbf{v}_{0,i} - \mathbf{v}_{0,j})^2$ (where –for simplicity– we consider the case in which the underlying geometry is one-dimensional). In principal component analysis (PCA) [15], one is given a data matrix $\mathbf{X} \in \mathbb{R}^{n \times p}$, and tries to reduce its dimensionality by postulating an approximate factorization $\mathbf{X} \approx \mathbf{u_0} \, \mathbf{v_0}^\mathsf{T}$. Hence $\mathbf{X}_{ij}$ can be interpreted as a noisy observation of the quadratic function $\mathbf{u}_{0,i} \mathbf{v}_{0,j}$. As a last example, there has been significant interest recently in phase retrieval problems [11, 6]. In this case, the unknown vector $\mathbf{v_0}$ is –roughly speaking– an image, and the observations are proportional to the square modulus of a modulated Fourier transform $|\mathbf{Fv_0}|^2$.

In several of these contexts, a significant effort has been devoted to exploiting additional structure of the unknown vector $\mathbf{v_0}$. For instance, in Sparse PCA, various methods have been developed to exploit the fact that $\mathbf{v_0}$ is known to be sparse [14, 25]. In sparse phase retrieval [13, 18], a similar assumption is made in the context of phase retrieval.

All of these attempts face a recurring dichotomy. One the hand, additional information on $\mathbf{v_0}$ can increase dramatically the estimation accuracy. On the other, only a fraction of this additional information is exploited by existing polynomial time algorithms. For instance in sparse PCA, if it is known that only $k$ entries of the vector $\mathbf{v_0}$ are non-vanishing, an optimal estimator is successful in identifying them from roughly $k$ samples (neglecting logarithmic factors) [2]. On the other hand, known polynomial-time algorithms require about $k^2$ samples [16, 7].

This fascinating phenomenon is however poorly understood so far. Classifying estimation problems as to whether optimal estimation accuracy can be achieved or not in polynomial time is an outstanding challenge. In this paper we develop a stylized model to study estimation from quadratic observations, under additional constraints. Special choices of the constraint set yield examples for which optimal estimation is thought to be intractable.

However we identify a large class of constraints for which estimation appears to be tractable, despite the corresponding maximum likelihood problem is non-convex. This shows that computational tractability is not immediately related to simple considerations of convexity or worst-case complexity.

Our model assumes $\mathbf{v_0} \in \mathcal{C}_n$ with $\mathcal{C}_n \subseteq \mathbb{R}^n$ a closed cone. Observations are organized in a symmetric matrix $\mathbf{X} = (\mathbf{X}_{ij})_{1 \leq i,j \leq n}$ defined by

$$\mathbf{X} = \beta \, \mathbf{v_0}\mathbf{v_0}^\mathsf{T} + \mathbf{Z} \,. \tag{1}$$

Here $\mathbf{Z}$ is a symmetric noise matrix with independent entries $(\mathbf{Z}_{ij})_{i \leq j}$ with $\mathbf{Z}_{ij} \sim \mathsf{N}(0, 1/n)$ for $i < j$ and $\mathbf{Z}_{ii} \sim \mathsf{N}(0, 2/n)$. We assume, without loss of generality, $\|\mathbf{v_0}\|_2 = 1$, and hence $\beta$ is the signal to noise ratio. We will assume $\beta$ to be known to avoid non-essential complications.

We consider estimators that return normalized vectors $\widehat{\mathbf{v}} : \mathbb{R}^{n \times n} \to \mathbb{S}^{n-1} \equiv \{\mathbf{v} \in \mathbb{R}^n : \|\mathbf{v}\|_2 = 1\}$, and will characterize such an estimator through the risk function

$$\mathsf{R}_{\mathcal{C}_n}(\widehat{\mathbf{v}}; \mathbf{v_0}) = \frac{1}{2}\mathbb{E}\big\{ \min(\|\widehat{\mathbf{v}}(\mathbf{X}) - \mathbf{v_0}\|_2^2, \|\widehat{\mathbf{v}}(\mathbf{X}) - \mathbf{v_0}\|_2^2)\big\} = 1 - \mathbb{E}\{|\langle\widehat{\mathbf{v}}(\mathbf{X}), \mathbf{v_0}\rangle|\} \,. \tag{2}$$

The corresponding worst-case risk is $\mathsf{R}(\widehat{\mathbf{v}}; \mathcal{C}_n) \equiv \sup_{\mathbf{v_0} \in \mathcal{C}_n} \mathsf{R}_{\mathcal{C}_n}(\widehat{\mathbf{v}}; \mathbf{v_0})$, and the minimax risk $\mathsf{R}(\mathcal{C}_n) = \inf_{\widehat{\mathbf{v}}} \mathsf{R}(\widehat{\mathbf{v}}; \mathcal{C}_n)$.

**Remark 1.1.** Let $\mathcal{C}_n = \mathcal{S}_{n,k}$ be the cone of vectors $\mathbf{v}_0$ that have at most $k$ non-zero entries, all positive, and with equal magnitude. The problem of testing whether $\beta = 0$ or $\beta \geq \beta_0$ within the model (1) coincides with the problem of detecting a non-zero mean submatrix in a Gaussian matrix. For the latter, Ma and Wu [20] proved that it cannot be accomplished in polynomial time unless an algorithm exists for the so-called planted clique problem in a regime in which the latter is conjectured to be hard.

This suggests that the problem of estimating $\mathbf{v_0}$ with rate-optimal minimax risk is hard for the constraint set $\mathcal{C}_n = \mathcal{S}_{n,k}$.

We next summarize our results. While –as shown by the last remark– optimal estimation is generically intractable for the model (1) under the constraint $\mathbf{v_0} \in \mathcal{C}_n$, we show that –roughly speaking– *it is instead tractable if $\mathcal{C}_n$ is a convex cone*. Note that this does not follow from elementary convexity considerations. Indeed, the maximum likelihood problem

$$\begin{aligned} \text{maximize} \quad & \langle\mathbf{v}, \mathbf{X}\mathbf{v}\rangle \,, \\ \text{subject to} \quad & \mathbf{v} \in \mathcal{C}_n, \quad \|\mathbf{v}\|_2 = 1 \,, \end{aligned} \tag{3}$$

is non-convex. Even more, solving exactly this optimization problem is NP-hard even for simple choices of the convex cone $\mathcal{C}_n$. For instance, if $\mathcal{C}_n = \mathcal{P}_n \equiv \{\mathbf{v} \in \mathbb{R}^n : \mathbf{v} \geq 0\}$ is an orthant, then solving the above is equivalent to copositive programming, which is NP-hard by reduction from maximum independent sets [12, Chapter 7].

Our results naturally characterize the cone $\mathcal{C}_n$ through its statistical dimension [1]. If $\mathsf{P}_{\mathcal{C}_n}$ denotes the orthogonal projection on $\mathcal{C}_n$, then the fractional statistical dimension of $\mathcal{C}_n$ is defined as

$$\delta(\mathcal{C}_n) \equiv \frac{1}{n}\mathbb{E}\big\{\big\|\mathsf{P}_{\mathcal{C}_n}(\mathbf{g})\big\|_2^2\big\} \,, \tag{4}$$

where expectation is with respect to $\mathbf{g} \sim \mathsf{N}(0, \mathsf{I}_{n \times n})$. Note that $\delta(\mathcal{C}_n) \in [0, 1]$ can be *significantly smaller* than 1. For instance, if $\mathcal{C}_n = \mathcal{M}_n \equiv \{\mathbf{v} \in \mathbb{R}_+^n : \forall i, \, \mathbf{v}_{i+1} \geq \mathbf{v}_i\}$ is the cone of non-negative, monotone increasing sequences, then [9, Lemma 4.2] proves that $\delta(\mathcal{C}_n) \leq 20(\log n)^2/n$.

Below is an informal summary of our results, with titles referring to sections where these are established.

**Information-theoretic limits.** We prove that in order to estimate accurately $\mathbf{v_0}$, it is necessary to have $\beta \gtrsim \sqrt{\delta(\mathcal{C}_n)}$. Namely, there exist universal constants $c_1, c_2 > 0$ such that, if $\beta \leq c_1 \sqrt{\delta(\mathcal{C}_n)}$, then $\mathsf{R}(\mathcal{C}_n) \geq c_2$.

**Maximum likelihood estimator.** Let $\widehat{\mathbf{v}}^{\mathrm{ML}}(\mathbf{X})$ be the maximum-likelihood estimator, i.e. any solution of Eq. (3). We then prove that, for $\beta \geq \sqrt{\delta(\mathcal{C}_n)}$

$$\mathsf{R}(\widehat{\mathbf{v}}^{\mathrm{ML}}; \mathcal{C}_n) \leq \frac{4\sqrt{\delta(\mathcal{C}_n)}}{\beta} \,. \tag{5}$$

**Low-complexity iterative estimator.** In the special case $\mathcal{C}_n = \mathbb{R}^n$, the solution of the optimization problem (3) is given by the eigenvector with the largest eigenvalue. A standard low-complexity approach to computing the leading eigenvector is provided by the power method. We consider a simple generalization that –starting from the initialization $\mathbf{v}^0$– alternates between projection onto $\mathcal{C}_n$ and multiplication by $(\mathbf{X} + \rho\mathrm{I}_n)$ ($\rho > 0$ is added to improve convergence):

$$\widehat{\mathbf{v}}^{t+1} = \frac{\mathsf{P}_{\mathcal{C}_n}(\mathbf{u}^t)}{\|\mathsf{P}_{\mathcal{C}_n}(\mathbf{u}^t)\|_2} \,, \tag{6}$$

$$\mathbf{u}^t = (\mathbf{X} + \rho\mathrm{I}_n)\widehat{\mathbf{v}}^t \,. \tag{7}$$

We prove that, for $t \gtrsim \log n$ iterations, this algorithm yields an estimate with $\mathsf{R}(\widehat{\mathbf{v}}^t; \mathcal{C}_n) \lesssim \sqrt{\delta(\mathcal{C}_n)}/\beta$, and hence order optimal, for $\beta \gtrsim \sqrt{\delta(\mathcal{C}_n)}$. (Our proof technique requires the initialization to have a positive scalar product with $\mathbf{v_0}$.)

As a side result of our analysis of the maximum likelihood estimator, we prove a new, elegant, upper bound on the value of the optimization problem (3), denoted by $\lambda_1(\mathbf{Z}; \mathcal{C}_n) \equiv \max_{\mathbf{v} \in \mathcal{C}_n \cap \mathbb{S}^{n-1}} \langle \mathbf{v}, \mathbf{Z}\mathbf{v} \rangle$. Namely

$$\mathbb{E}\lambda_1(\mathbf{Z}; \mathcal{C}_n) \leq 2\sqrt{\delta(\mathcal{C}_n)} \,. \tag{8}$$

In the special case $\mathcal{C}_n = \mathbb{R}^n$, $\lambda_1(\mathbf{Z}; \mathbb{R}^n)$ is the largest eigenvalue of $\mathbf{Z}$, and the above inequality shows that this is bounded in expectation by 2. In this case, the bound is known to be asymptotically tight [10]. In the supplementary material, we prove that it is tight for certain other examples such as the nonnegative orthant and for circular cones (a.k.a. ice-cream cones). We conjecture that this inequality is *asymptotically tight for general convex cones*.

Unless stated otherwise, in the following we will defer proofs to the Supplementary Material.

## 2 Information-theoretic limits

We use an information-theoretic argument to show that, under the observation model (1), then the minimax risk can be bounded below for $\beta \lesssim \sqrt{\delta(\mathcal{C}_n)}$. As is standard, our bound employs the so-called packing number of $\mathcal{C}_n$.

**Definition 2.1.** *For a cone $\mathcal{C}_n \subseteq \mathbb{R}^n$, we define its packing number $N(\mathcal{C}_n, \varepsilon)$ as the size of the maximal subset $\mathcal{X}$ of $\mathcal{C}_n \cap \mathbb{S}^{n-1}$ such that for every $x_1, x_2 \in \mathcal{C}_n \cap \mathbb{S}^{n-1}$, $\|x_1 - x_2\| \geq \varepsilon$.*

We then have the following.

**Theorem 1.** *There exist universal constants $C_1, C_2 > 0$ such that for any closed convex cone $\mathcal{C}_n$ with $\delta(\mathcal{C}_n) \geq 3/n$:*

$$\beta \leq C_1\sqrt{\delta(\mathcal{C}_n)} \quad \Rightarrow \quad \mathsf{R}(\mathcal{C}_n) \geq \frac{C_2\delta(\mathcal{C}_n)}{\log(1/\delta(\mathcal{C}_n))} \,. \tag{9}$$

Notice that the last expression for the lower bound depends on the cone width, as it is to be expected: even for $\beta = 0$, it is possible to estimate $\mathbf{v_0}$ with risk going to 0 if the cone $\mathcal{C}_n$ 'shrinks' as $n \to \infty$. The proof of this theorem is provided in Section 2 of the supplement.

## 3 Maximum likelihood estimator

Under the Gaussian noise model for $\mathbf{Z}$, cf. Eq. (1), the likelihood of observing $\mathbf{X}$ under a hypothesis $\mathbf{v}$ is proportional to $\exp(-\|\mathbf{X} - \mathbf{v}\mathbf{v}^{\mathsf{T}}\|_F^2/2)$. Using the constraint that $\|\mathbf{v}\| = 1$, it follows that any solution of (3) is a maximum likelihood estimator.

**Theorem 2.** *Consider the model as in* (1). *Then, when* $\beta \geq \sqrt{\delta(\mathcal{C}_n)}$, *any solution* $\widehat{\mathbf{v}}^{\mathrm{ML}}(\mathbf{X})$ *to the maximum likelihood problem* (3) *satisfies*

$$\mathsf{R}_{\mathcal{C}_n}(\widehat{\mathbf{v}}^{\mathrm{ML}}; \mathcal{C}_n) \leq \min\left\{\frac{4\sqrt{\delta(\mathcal{C}_n)}}{\beta}, \frac{16}{\beta^2}\right\}. \tag{10}$$

Thus, for $\beta \gtrsim \sqrt{\delta(\mathcal{C}_n)}$, the risk of the maximum likelihood estimator decays as $\sqrt{\delta(\mathcal{C}_n)}/\beta$ while for $\beta \gtrsim 1$, it shifts to a faster decay of $1/\beta^2$. We have made no attempt to optimize the constants in the statement of the theorem, though we believe that the correct leading constant in either case is 1.

Note that without the cone constraint (or with $\mathcal{C}_n = \mathbb{R}^n$) the maximum likelihood estimator reduces to computing the principal eigenvector $\widehat{\mathbf{v}}^{\mathrm{PC}}$ of $\mathbf{X}$. Recent results in random matrix theory [10] and statistical decision theory [4] prove that in the case of principal eigenvector, a nontrivial risk (i.e. $\mathsf{R}_{\mathcal{C}_n}(\widehat{\mathbf{v}}^{\mathrm{PC}}; \mathcal{C}_n) < 1$ asymptotically) is obtained only when $\beta > 1$. Our result shows that this threshold is, instead, reduced to $\sqrt{\delta(\mathcal{C}_n)}$, which can be significantly smaller than 1. The proof of this theorem is provided in Section 3 of the supplement.

## 4 Low-complexity iterative estimator

Sections 2 and 3 provide theoretical insight into the fundamental limits of estimation of $\mathbf{v_0}$ from quadratic observations of the form $\beta \mathbf{v_0}\mathbf{v_0}^\mathsf{T} + \mathbf{Z}$. However, as previously mentioned, the maximum likelihood estimator of Section 3 is NP-hard to compute, in general. In this section, we propose a simple iterative algorithm that generalizes the well-known power iteration to compute the principal eigenvector of a matrix. Furthermore, we prove that, given an initialization with positive scalar product with $\mathbf{v_0}$, this algorithm achieves the same risk of the maximum likelihood estimator up to constants. Throughout, the cone $\mathcal{C}_n$ is assumed to be convex.

Our starting point is the power iteration to compute the principal eigenvector $\widehat{\mathbf{v}}^{\mathrm{PC}}$ of $\mathbf{X}$. This is given by letting, for $t \geq 0$: $\widehat{\mathbf{v}}^{t+1} = \mathbf{X}\widehat{\mathbf{v}}^t/\|\mathbf{X}\widehat{\mathbf{v}}^t\|$. Under our observation model, we have $\mathbf{X} = \beta \mathbf{v_0}\mathbf{v_0}^\mathsf{T} + \mathbf{Z}$ with $\mathbf{v_0} \in \mathcal{C}_n$. We can incorporate this information by projecting the iterates on to the cone $\mathcal{C}_n$ (see e.g. [19] for related ideas):

$$\widehat{\mathbf{v}}^t = \frac{\mathsf{P}_{\mathcal{C}_n}(\mathbf{u}^t)}{\|\mathsf{P}_{\mathcal{C}_n}(\mathbf{u}^t)\|}, \qquad \mathbf{u}^{t+1} = \mathbf{X}\mathbf{v}^t + \rho \mathbf{v}^t. \tag{11}$$

The projection is defined in the standard way:

$$\mathsf{P}_{\mathcal{C}_n}(\mathbf{x}) \equiv \arg\min_{\mathbf{y} \in \mathcal{C}_n} \|\mathbf{y} - \mathbf{x}\|^2. \tag{12}$$

If $\mathcal{C}_n$ is convex, then the projection is unique. We have implicitly assumed that the operation of projecting to the cone $\mathcal{C}_n$ is available to the algorithm as a simple primitive. This is the case for many convex cones of interest, such as the orthant $\mathcal{P}_n$, the monotone cone $\mathcal{M}_n$, and ice-cream cones the projection is easy to compute. For instance, if $\mathcal{C}_n = \mathcal{P}_n$ is the non-negative orthant $\mathsf{P}_{\mathcal{C}_n}(\mathbf{x}) = (\mathbf{x})_+$ is the non-negative part of $\mathbf{x}$. For the monotone cone, the projection can be computed efficiently through the pool-adjacent violators algorithm.

The memory term $\rho \mathbf{v}^t$ is necessary for our proof technique to go through. It is straightforward to see that adding $\rho \mathsf{I}_n$ to the data $\mathbf{X}$ does not change the optimizers of the problem (3). The following theorem provides deterministic conditions under which the distance between the iterative estimator and the vector $\mathbf{v_0}$ can be bounded.

**Theorem 3.** *Let* $\widehat{\mathbf{v}}^t$ *be the power iteration estimator* (11). *Assume* $\rho > \Delta$ *and that the noise matrix* $\mathbf{Z}$ *satisfies:*

$$\max\left\{|\langle \mathbf{x}, \mathbf{Z}\mathbf{y}\rangle| : \mathbf{x}, \mathbf{y} \in \mathcal{C}_n \cap \mathbb{S}^{n-1}\right\} \leq \Delta. \tag{13}$$

*If* $\beta > 4\Delta$, *and the initial point* $\widehat{\mathbf{v}}^0 \in \mathcal{C}_n \cap \mathbb{S}^{n-1}$ *satisfies* $\langle \widehat{\mathbf{v}}^0, \mathbf{v_0}\rangle \geq 2\Delta/\beta$, *then there exits* $t_0 = t_0(\Delta/\beta, \Delta/\rho) < \infty$ *independent of* $n$ *such that, for all* $t \geq t_0$

$$\|\widehat{\mathbf{v}}^t - \mathbf{v_0}\| \leq \frac{4\Delta}{\beta}. \tag{14}$$

We can apply this theorem to the Gaussian noise model to obtain the following bound on the risk of the power iteration estimator.

**Corollary 4.1.** *Under the model* (1) *let* $\varepsilon_n = 8\sqrt{\log n/n}$. *Assume that* $\langle \widehat{\mathbf{v}}^0, \mathbf{v_0} \rangle > 0$ *and*

$$\beta > 2(\sqrt{\delta(\mathcal{C}_n)} + \varepsilon_n) \, \max\left(2, \, \langle \widehat{\mathbf{v}}^0, \mathbf{v_0} \rangle^{-1}\right). \tag{15}$$

$$\textit{Then} \quad \mathsf{R}(\widehat{\mathbf{v}}^t, \mathcal{C}_n) \leq \frac{2\delta(\mathcal{C}_n) + \varepsilon_n}{\beta}. \tag{16}$$

In other words, power iteration has risk within a constant from the maximum likelihood estimator, provided an initialization is available whose scalar product with $\mathbf{v_0}$ is bounded away from zero. The proofs of Theorem 3 and Corollary 4.1 are provided in Section 4 of the supplement.

## 5  A case study: sharp asymptotics and minimax results for the orthant

In this section, we will be interested in the example in which the cone $\mathcal{C}_n$ is the non-negative orthant $\mathcal{C}_n = \mathcal{P}_n$. Non-negativity constraints within principal component analysis arise in non-negative matrix factorization (NMF). Initially introduced in the context of chemometrics [23, 22], NMF attracted considerable interest because of its applications in computer vision and topic modeling. In particular, Lee and Seung [17] demonstrated empirically that NMF successfully identifies parts of images, or topics in documents' corpora.

Note that the in applications of NMF to computer vision or topic modeling the setting is somewhat different from the model studied here: $\mathbf{X}$ is rectangular instead of symmetric, and the rank is larger than one. Such generalizations of our analysis will be the object of future work.

Here we will use the positive orthant to illustrate the results in previous sections. Further, we will show that stronger results can be proved in this case, thanks to the separable structure of this cone. Namely, we derive sharp asymptotics and we characterize the least-favorable vectors for the maximum likelihood estimator.

We denote by $\lambda^+(\mathbf{X}) = \lambda_1(\mathbf{X}; \mathcal{C}_n = \mathcal{P}_n)$ the value of the optimization problem (3). Our first result yields the asymptotic value of this quantity for 'pure noise,' confirming the general conjecture put forward above.

**Theorem 4.** *We have almost surely* $\lim_{n\to\infty} \lambda^+(\mathbf{Z}) = 2\sqrt{\delta(\mathcal{P}_n)} = \sqrt{2}$.

Next we characterize the risk phase transition: this result confirms and strengthen Theorem 2.

**Theorem 5.** *Consider estimation in the non-negative orthant* $\mathcal{C}_n = \mathcal{P}_n$ *under the model (1). If* $\beta \leq 1/\sqrt{2}$, *then there exists a sequence of vectors* $\{\mathbf{v}_0(n)\}_{n\geq 0}$, *such that almost surely*

$$\lim_{n\to\infty} \mathsf{R}(\mathbf{v}^{\mathrm{ML}}; \mathbf{v_0}(n)) = 1. \tag{17}$$

*For* $\beta > 1/\sqrt{2}$, *there exists a function* $\beta \mapsto R_+(\beta)$ *with* $R_+(\beta) < 1$ *for all* $\beta > 1/\sqrt{2}$, *and* $R_+(\beta) \geq 1 - 1/2\beta^2$, *such that the following happens. For any sequence of vectors* $\{\mathbf{v}_0(n)\}_{n\geq 0}$, *we have, almost surely*

$$\limsup_{n\to\infty} \mathsf{R}(\mathbf{v}^{\mathrm{ML}}; \mathbf{v_0}(n)) \leq R_+(\beta). \tag{18}$$

In other words, in the high-dimensional limit, the maximum likelihood estimator is positively correlated with the signal $\mathbf{v_0}(n)$ if and only if $\beta > \sqrt{\delta(\mathcal{C}_n)} = 1/\sqrt{2}$.

Explicit (although non-elementary) expressions for $R_+(\beta)$ can be computed, along with the limit value of the risk $\mathsf{R}(\mathbf{v}^{\mathrm{ML}}; \mathbf{v_0}(n))$ for sequences of vectors $\{\mathbf{v}_0(n)\}_{n\geq 1}$ whose entries empirical distribution converges. These results go beyond the scope of the present paper (but see Fig. 1 below for illustration).

As a byproduct of our analysis, we can characterize the least-favorable choice of the signal $\mathbf{v_0}$. Namely for $k \in [1, n]$, wee let $\mathbf{u}(n, k)$ denote a vector with $\lfloor k \rfloor$ non-zero entries, all equal to $1/\sqrt{\lfloor k \rfloor}$. Then we can prove that the asymptotic minimax risk is achieved along sequences of vectors of this type.

**Theorem 6.** *Consider estimation in the non-negative orthant $\mathcal{C}_n = \mathcal{P}_n$ under the model (1), and let $R_+(\beta)$ be the same function as in Theorem 5. If $\beta \le 1/\sqrt{2}$ then there exists $k_n = o(n)$ such that*

$$\lim_{n\to\infty} \mathsf{R}(\mathbf{v}^{\mathrm{ML}}; \mathbf{u}(n, k_n)) = 1 \,. \tag{19}$$

*If $\beta > 1/\sqrt{2}$ then there exists $\varepsilon_\# = \varepsilon_\#(\beta) \in (0, 1]$ such that*

$$\lim_{n\to\infty} \mathsf{R}(\mathbf{v}^{\mathrm{ML}}; \mathbf{u}(n, n\varepsilon_\#)) = R_+(\beta) \,. \tag{20}$$

We refer the reader to [21] for a detailed analysis of the case of nonnegative PCA and the full proofs of Theorems 4, 5 and 6.

## 5.1 Approximate Message Passing

The next question is whether, in the present example $\mathcal{C}_n = \mathcal{P}_n$, the risk of the maximum likelihood estimator can be achieved by a low-complexity iterative algorithm. We prove that this is indeed the case (up to an arbitrarily small error), thus confirming Theorem 3. In order to derive an asymptotically exact analysis, we consider an 'approximate message passing' modification of the power iteration.

Let $f(\mathbf{x}) = (\mathbf{x})_+ / \|(\mathbf{x})_+\|_2$ denote the normalized projector. We consider the iteration defined by $\mathbf{v}^0 = (1, 1, \dots, 1)^{\mathsf{T}} / \sqrt{n}$, $\mathbf{v}^{-1} = (0, 0, \dots, 0)^{\mathsf{T}}$, and for $t \ge 0$,

$$\mathbf{v}^{t+1} = \mathbf{X} f(\mathbf{v}^t) - \mathsf{b}_t \, f(\mathbf{v}^{t-1}) \quad \text{and} \quad \mathsf{b}_t \equiv \|(\mathbf{v}^t)_+\|_0 / \{\sqrt{n} \|(\mathbf{v}^t)_+\|_2\} \qquad \text{AMP}$$

The algorithm AMP is a slight modification of the projected power iteration algorithm up to adding at each step the "memory term" $-\mathsf{b}_t \, f(\mathbf{v}^{t-1})$. As shown in [8, 3] this term plays a crucial role in allowing for an exact high-dimensional characterization. At each step the estimate produced by the sequence is $\widehat{\mathbf{v}}^t = (\mathbf{v}^t)_+ / \|(\mathbf{v}^t)_+\|_2$. We have the following

**Theorem 7.** *Let $\mathbf{X}$ be generated as in (1). Then we have, almost surely,*

$$\lim_{t\to\infty} \lim_{n\to\infty} \left| \langle \widehat{\mathbf{v}}^{\mathrm{ML}}, \mathbf{X}\mathbf{v}^{\mathrm{ML}} \rangle - \langle \widehat{\mathbf{v}}^t, \mathbf{X}\widehat{\mathbf{v}}^t \rangle \right| = 0 \ . \tag{21}$$

## 5.2 Numerical illustration: comparison with classical PCA

We performed numerical experiments on synthetic data generated according to the model (1) and with signal $\mathbf{v_0} = \mathbf{u}(n, n\varepsilon)$ as defined in the previous section. We provide in the Appendix formulas for the value of $\lim_{n\to\infty} \langle \mathbf{v_0}, \widehat{\mathbf{v}}^{\mathrm{ML}} \rangle$, which correspond to continuous black lines in the Figure 1. We compare these predictions with empirical values obtained by running AMP.

We generated samples of size $n = 10^4$, sparsity level $\varepsilon \in \{0.001, 0.1, 0.8\}$, and signal-to-noise ratios $\beta \in \{0.05, 0.10, \dots, 1.5\}$. In each case we run AMP for $t = 50$ iterations and plot the empirical average of $\langle \widehat{\mathbf{v}}^t, \mathbf{v_0} \rangle$ over 32 instances. Even for such moderate values of $n$, the asymptotic predictions are remarkably accurate.

Observe that sparse vectors (small $\varepsilon$) correspond to the least favorable signal for small signal-to-noise ratio $\beta$, while the situation is reverted for large values of $\beta$. In dashed green we represented the theoretical prediction for $\varepsilon \to 0$. The value $\beta = 1/\sqrt{2}$ corresponds to the phase transition. At the bottom the images correspond to values of the correlation $\langle \mathbf{v_0}, \widehat{\mathbf{v}}^{\mathrm{ML}} \rangle$ for a grid of values of $\beta$ and $\varepsilon$. The top left-hand frame in Figure 1 is obtained by repeating the experiment for a grid of values of $n$, and fixed $\varepsilon = 0.05$ and several value of $\beta$. For each point we plot the average of $\langle \widehat{\mathbf{v}}^t, \mathbf{v_0} \rangle$ after $t = 50$ iteration, over 32 instances. The data suggest $\langle \widehat{\mathbf{v}}^{\mathrm{ML}}, \mathbf{v_0} \rangle + A \, n^{-b} \approx \lim_{n\to\infty} \langle \mathbf{v_0}, \mathbf{v_+} \rangle$ with $b \approx 0.5$.

## 6 Polyhedral cones and convex relaxations

A polyhedral cone $\mathcal{C}_n$ is a closed convex cone that can be represented in the form $\mathcal{C}_n = \{\mathbf{x} \in \mathbb{R}^n : \mathbf{A}\mathbf{x} \ge 0\}$ for some matrix $\mathbf{A} \in \mathbb{R}^{m \times n}$. In section 5 we considered the non-negative orthant, which is an example of polyhedral cone with $\mathbf{A} = \mathbf{I}_n$. A number of other examples of practical interest fall within this category of cones. For instance, monotonicity or convexity of a vector $\mathbf{v} = (\mathbf{v}_1, \dots, \mathbf{v}_n)$

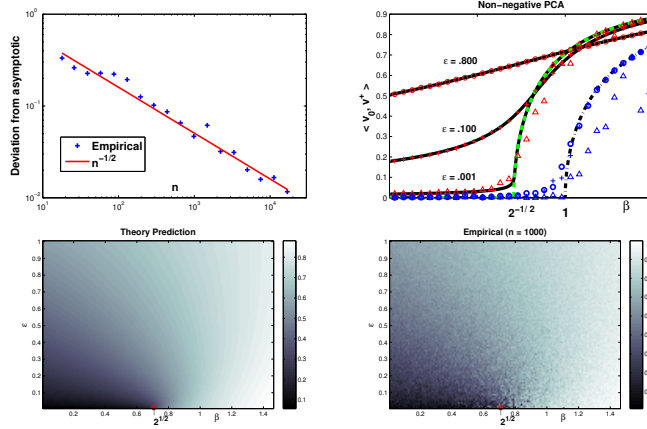

Figure 1: Numerical simulations with the model 1 for the positive orthant cone $\mathcal{C}_n = \mathcal{P}_n$. Top-left: empirical deviation from asymptotic prediction. Top-right: black lines represent the theoretical predictions of Theorem 5, and dots represent empirical values of $\langle \widehat{\mathbf{v}}^t, \mathbf{v_0} \rangle$ for the AMP estimator (in red) and $\langle \mathbf{v}_1, \mathbf{v_0} \rangle$ for standard PCA (in blue). Bottom: a comparison of theoretical asymptotic values (left frame) and empirical values (right frame) of $\langle \mathbf{v_0}, \mathbf{v}^{\mathrm{ML}} \rangle$ for a range of $\beta$ and $\varepsilon$.

an be enforced –in their discrete version– through inequality constraints (respectively $\mathbf{v}_{i+1} - \mathbf{v}_i \geq 0$ and $\mathbf{v}_{i+1} - 2\mathbf{v}_i + \mathbf{v}_{i-1} \geq 0$), and hence give rise to polyhedral cones. Furthermore, it is possible to approximate any convex cone $\mathcal{C}_n$ with a sequence of increasingly accurate polyhedral cones.

For a polyhedral cone, the maximum likelihood problem (3) reads:

$$\text{maximize } \langle \mathbf{v}, \mathbf{X}\mathbf{v} \rangle \tag{22}$$
$$\text{subject to: } \mathbf{A}\mathbf{v} \geq 0; \|\mathbf{v}\| = 1.$$

The modified power iteration (11), can be specialized to this case, via the appropriate projection. The projection remains computationally feasible provided the matrix $\mathbf{A}$ is not too large. Indeed, it is easy to show using convex duality that $\mathsf{P}_{\mathcal{C}_n}(\mathbf{u})$ is given by:

$$\mathsf{P}_{\mathcal{C}_n}(\mathbf{u}) = \arg\min \left\{ \|\mathbf{A}\mathbf{x} + \mathbf{u}\|^2, \mathbf{x} \geq 0 \right\}.$$

This reduces the projection onto a general polyhedral cone to a non-negative least squares problem, for which efficient routines exist. In special cases such as the orthant, the projection is closed form. In the case of polyhedral cones, it is possible to relax this problem (22) using a natural convex surrogate. To see this, we introduce the variable $\mathbf{V} = \mathbf{v}\mathbf{v}^\mathsf{T}$ and write the following *equivalent* version of problem 22:

$$\text{maximize } \langle \mathbf{X}, \mathbf{V} \rangle$$
$$\text{subject to: } \mathbf{A}\mathbf{V}\mathbf{A}^\mathsf{T} \geq 0; \mathrm{Tr}(\mathbf{V}) = 1;$$
$$\mathbf{V} \succeq 0; \mathrm{rank}(\mathbf{V}) = 1.$$

Here the constraint $\mathbf{A}\mathbf{V}\mathbf{A}^\mathsf{T} \geq 0$ is to be interpreted as entry-wise non-negativity, while we write $\mathbf{V} \succeq 0$ to denote that $\mathbf{V}$ is positive semidefinite. We can now relax this problem by dropping the rank constraint:

$$\text{maximize } \langle \mathbf{X}, \mathbf{V} \rangle \tag{23}$$
$$\text{subject to: } \mathbf{A}\mathbf{V}\mathbf{A}^\mathsf{T} \geq 0; \mathrm{Tr}(\mathbf{V}) = 1; \mathbf{V} \succeq 0.$$

Note that this increases the number of variables from $n$ to $n^2$, as $\mathbf{V} \in \mathbb{R}^{n \times n}$, which results in a significant cost increase for standard interior point methods, over the power iteration (11). Furthermore, if the solution $\mathbf{V}$ is *not* rank one, it is not clear how one can use it to form an estimate $\widehat{\mathbf{v}}$. On the other hand, this convex relaxation yields a principled approach to bounding the sub-optimality

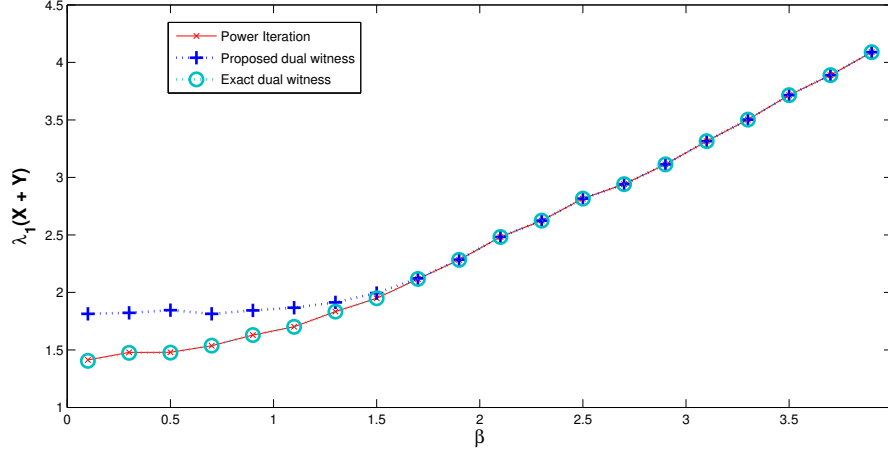

Figure 2: Value of the maximum likelihood problem (3) for $\mathcal{C}_n = \mathcal{P}_n$, as approximated by power iteration. The red line is the value achieved by power iteration, and the blue points the upper bound obtained by dual witness (25). The gap at small $\beta$ is due to the suboptimal choice of the dual witness, since solving exactly Problem (24) yields the dual witness with value given by the teal circles. As can be seen, they match exactly the value obtained by power iteration, showing zero duality gap! The simulation is for $n = 50$ and 40 Monte Carlo iterations.

of the estimate provided by the power iteration. It is straightforward to derive the dual program of (23):

$$\text{minimize } \lambda_1(\mathbf{X} + \mathbf{A}^\mathsf{T}\mathbf{Y}\mathbf{A}) \tag{24}$$
$$\text{subject to: } \mathbf{Y} \geq 0,$$

where $\mathbf{Y}$ is the decision variable, the constraint is interpreted as entry-wise nonnegativity as above, and $\lambda_1(\,\cdot\,)$ denotes the largest eigenvalue. If one can construct a *dual witness* $\mathbf{Y} \geq 0$ such that $\lambda_1(\mathbf{X} + \mathbf{A}^\mathsf{T}\mathbf{Y}\mathbf{A}) = \langle \widehat{\mathbf{v}}, \mathbf{X}\widehat{\mathbf{v}} \rangle$ for any estimator $\widehat{\mathbf{v}}$, then this estimator is the maximum likelihood estimator. In particular, using the power iteration estimator $\widehat{\mathbf{v}} = \widehat{\mathbf{v}}^t$, such a dual witness can provide a *certificate of convergence* of the power iteration (11).

We next describe a construction of dual witness that we found empirically successful at large enough signal-to-noise ratio. Assume that a heuristic (for instance, the modified power iteration (11)) has produced an estimate $\widehat{\mathbf{v}}$ that is a local maximizer of the problem (3). It is is proved in the Supplementary Material, that such a local maximizer must satisfy the modified eigenvalue equation: $\mathbf{X}\widehat{\mathbf{v}} = \lambda\widehat{\mathbf{v}} - \mathbf{A}^\mathsf{T}\boldsymbol{\mu}$, with $\boldsymbol{\mu} \geq 0$ and $\langle \widehat{\mathbf{v}}, \mathbf{A}^\mathsf{T}\boldsymbol{\mu} \rangle = 0$.

We then suggest the witness

$$\mathbf{Y}(\widehat{\mathbf{v}}) = \frac{1}{\|\mathbf{A}\widehat{\mathbf{v}}\|^2}\left(\boldsymbol{\mu}\widehat{\mathbf{v}}^\mathsf{T}\mathbf{A}^\mathsf{T} + \mathbf{A}\widehat{\mathbf{v}}\boldsymbol{\mu}^\mathsf{T}\right). \tag{25}$$

Note that $\mathbf{Y}(\widehat{\mathbf{v}})$ is non-negative by construction and hence dual feasible. A direct calculation shows that $\widehat{\mathbf{v}}$ is an eigenvector of the matrix $\mathbf{X} + \mathbf{A}^\mathsf{T}\mathbf{Y}\mathbf{A}$ with eigenvalue $\lambda = \langle \widehat{\mathbf{v}}, \mathbf{X}\widehat{\mathbf{v}} \rangle$. We then obtain the following sufficient condition for optimality.

**Proposition 6.1.** *Let $\widehat{\mathbf{v}}$ be a local maximizer of the problem* (3)*. If $\widehat{\mathbf{v}}$ is the principal eigenvector of* $\mathbf{X} + \mathbf{A}^\mathsf{T}\mathbf{Y}(\widehat{\mathbf{v}})\mathbf{A}$*, then $\widehat{\mathbf{v}}$ is a global maximizer.*

In Figure 2 we plot the average value of the objective function over 50 instances of the problem for $\mathcal{C}_n = \mathcal{P}_n$, $n = 100$. We solved the maximum likelihood problem using the power iteration heuristics (11), and used the above construction to compute an upper bound via duality. It is possible to show that this upper bound cannot be tight unless $\beta > 1$, but appears to be quite accurate. We also solve the problem (24) directly for case of nonnegative PCA, and (rather surprisingly) the dual is tight for every $\beta > 0$.

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
