[Supplementary Material]

# Supplement to "Cone-Constrained Principal Component Analysis"

**Yash Deshpande**
Electrical Engineering
Stanford University

**Andrea Montanari**
Electrical Engineering and Statistics
Stanford University

**Emile Richard**
Electrical Engineering
Stanford University

The supplement is organized as follows. In Section 1 we give an overview of the setting considered in the main article, the notation used and cover some useful general facts. In Sections 2, 3 and 4 we provide the proofs of Theorems 1 2 and 3 of the main article.

## 1   Preliminaries

Recall the setting in Problem (3) in the main article, we let $\mathbf{X} \in \mathbb{R}^{n \times n}$ be a symmetric matrix and $\mathcal{C}_n \subseteq \mathbb{R}^n$ be a closed cone (not necessarily convex). We consider the following optimization problem:

$$\text{maximize} \ \ \langle \mathbf{v}, \mathbf{X}\mathbf{v} \rangle, \tag{1}$$
$$\text{subject to} \ \ \mathbf{v} \in \mathcal{C}_n, \ \ \|\mathbf{v}\|_2 = 1.$$

We denote by $\lambda_{\max}(\mathbf{X}; \mathcal{C}_n)$ the value of this problem. For instance $\lambda_{\max}(\mathbf{X}; \mathbb{R}^n)$ is the largest eigenvalue of $\mathbf{X}$. Note that this optimization problem is –in general– NP-hard. We denote by $\mathcal{C}_n^*$ the dual cone of $\mathcal{C}_n$:

$$\mathcal{C}_n^* = \{\mathbf{x} \ : \ \forall \mathbf{z} \in \mathcal{C}, \ \langle \mathbf{x}, \mathbf{z} \rangle \geq 0\} \ ,$$

and by $\mathcal{C}_n^\circ$ its polar cone $\mathcal{C}_n^\circ = -\mathcal{C}_n^*$.

Further, we let $\mathcal{C}_{n,\mathbf{x}} = \mathrm{cone}\{\mathbf{y} - \mathbf{x} \ : \ \mathbf{y} \in \mathcal{C}_n\}$ be the tangent cone of $\mathcal{C}_n$ at $\mathbf{x} \in \mathcal{C}_n$ (in particular $\mathcal{C}_{n,0} = \mathcal{C}_n$), and $\mathcal{C}_{n,\mathbf{x}}^*$ its dual (i.e. $\mathcal{C}_{n,\mathbf{x}}^* = (\mathcal{C}_{n,\mathbf{x}})^*$). Finally, we let $\mathbb{S}^{n-1} = \{\mathbf{x} \ : \ \|\mathbf{x}\|_2 = 1\}$ be the unit sphere in $n$ dimensions.

Given a closed convex set $K \subseteq \mathbb{R}^n$, we let $\mathsf{P}_K : \mathbb{R}^n \to \mathbb{R}^n$ denote the orthogonal projection onto $K$. Given vectors $\mathbf{v}_1, \mathbf{v}_2, \ldots, \mathrm{span}(\mathbf{v}_1, \mathbf{v}_2, \ldots)$ denotes their linear span. For a linear subspace $V$, $V^\perp$ denotes its orthogonal complement.

The following definitions will be useful in the sequel.

**Definition 1.1.** *A* local maximum *of the problem (1) is a point* $\mathbf{v}^* \in \mathcal{C}_n \cap \mathbb{S}^{n-1}$, $\|\mathbf{v}^*\|_2 = 1$ *for which there exists an* $\varepsilon > 0$ *such that* $\langle \mathbf{v}, \mathbf{X}\mathbf{v} \rangle \leq \langle \mathbf{v}^*, \mathbf{X}\mathbf{v}^* \rangle$ *for all* $\mathbf{v} \in \mathcal{C}_n \cap \mathbb{S}^{n-1}$ *within a neighborhood of* $\mathbf{v}^*$: $\|\mathbf{v} - \mathbf{v}^*\|_2 \leq \delta$.

**Definition 1.2.** *Let* $\mathcal{C}_n$ *be a closed convex cone. For any* $\mathbf{x} \in \mathbb{R}^n$, *the Moreau decomposition of* $\mathbf{x}$ *is defined as* $\mathbf{x} = \mathsf{P}_{\mathcal{C}_n}(\mathbf{x}) + \mathsf{P}_{\mathcal{C}_n^\circ}(\mathbf{x})$. *Further we have that* $\langle \mathsf{P}_{\mathcal{C}_n}(\mathbf{x}), \mathsf{P}_{\mathcal{C}_n^\circ}(\mathbf{x}) \rangle = 0$.

### 1.1   General facts

Let $\mathbf{v}^*$ denote a local maximizer of (1). We begin with the following remark that characterizes the tangent cone at a local maximizer.

**Remark 1.3.** We have that $\langle \mathbf{v}^*, \mathbf{v} \rangle = 0$ for all $\mathbf{v} \in \mathcal{C}_{n,\mathbf{v}^*}^*$.

*Proof.* If $\mathbf{v}^* = 0$, the result is trivial. If $\mathbf{v}^* \neq 0$, we have that $\mathbf{v}^* - \mathbf{v}^* = 0 \in \mathcal{C}_n$ and $\mathbf{v}^* - (-\mathbf{v}^*)2\mathbf{v}^* \in \mathcal{C}_n$. Hence $\{\mathbf{v}^*, -\mathbf{v}^*\} \subseteq \mathcal{C}_{n,\mathbf{v}^*}$ and as a consequence for $\mathbf{v} \in \mathcal{C}_{n,\mathbf{v}^*}^*$, $\langle \mathbf{v}^*, \mathbf{v} \rangle \geq 0$ and $\langle -\mathbf{v}^*, \mathbf{v} \rangle \geq 0$ imply $\langle \mathbf{v}^*, \mathbf{v} \rangle = 0$. $\qquad \square$

The following proposition characterizes the value at the local maxima of Problem (1).

**Proposition 1.4.** *If the point* $\mathbf{v}^* \in \mathcal{C}_n \cap \mathbb{S}^{n-1}$ *is a local maximum of the problem (1), then there exists* $\boldsymbol{\mu}^* \in \mathcal{C}_{n,\mathbf{v}^*}^*$ *and* $\lambda^* \in \mathbb{R}$ *such that*

$$\mathbf{X}\mathbf{v}^* = \lambda^* \mathbf{v}^* - \boldsymbol{\mu}^*, \tag{2}$$

$$\lambda^* \geq \sup\left\{ \langle \mathbf{v}, \mathbf{X}\mathbf{v} \rangle : \mathbf{v} \in \mathcal{C}_{n,\mathbf{v}^*} \cap \operatorname{span}(\boldsymbol{\mu}^*)^\perp, \|\mathbf{v}\|_2 = 1 \right\}. \tag{3}$$

*Vice-versa, if the above conditions (2), (3) hold for some* $\mathbf{v}^* \in \mathcal{C}_n \cap \mathbb{S}^{n-1}$ *with the last inequality being strict, then* $\mathbf{v}^*$ *is a local maximum. Further, if* $\mathcal{C}_n$ *is convex Eq. (2) is the Moreau decomposition of* $\mathbf{X}\mathbf{v}^*$ *with respect to* $\mathcal{C}_n$. *In particular*

$$\lambda^* \mathbf{v}^* = \mathsf{P}_{\mathcal{C}_{n,\mathbf{v}^*}}(\mathbf{X}\mathbf{v}^*) = \mathsf{P}_{\mathcal{C}_n}(\mathbf{X}\mathbf{v}^*), \quad \boldsymbol{\mu}^* = -\mathsf{P}_{\mathcal{C}_{n,\mathbf{v}^*}^\circ}(\mathbf{X}\mathbf{v}^*) = -\mathsf{P}_{\mathcal{C}_n^\circ}(\mathbf{X}\mathbf{v}^*). \tag{4}$$

*Proof.* Note that, for any $\mathbf{v} \in \mathbf{relint}(\mathcal{C}_{n,\mathbf{v}^*})$ and $\varepsilon \geq 0$ we know that $\varepsilon \mathbf{v} \in \mathcal{C}_{n,\mathbf{v}^*}$. Therefore, by definition of the tangent cone, $\mathbf{v}(\varepsilon) \equiv \mathbf{v}^* + \varepsilon\,\mathbf{v} \in \mathcal{C}_n$. Then by local optimality of $\mathbf{v}^*$, for all $\varepsilon$ small enough we have

$$\langle \mathbf{v}(\varepsilon), \mathbf{X}\mathbf{v}(\varepsilon) \rangle \leq \langle \mathbf{v}^*, \mathbf{X}\mathbf{v}^* \rangle \, \|\mathbf{v}(\varepsilon)\|^2.$$

Expanding both sides and letting $\lambda^* = \langle \mathbf{v}^*, \mathbf{X}\mathbf{v}^* \rangle$, we get

$$2\varepsilon \langle \mathbf{v}, \mathbf{X}\mathbf{v}^* \rangle + \varepsilon^2 \langle \mathbf{v}, \mathbf{X}\mathbf{v} \rangle \leq 2\lambda^* \varepsilon \langle \mathbf{v}^*, \mathbf{v} \rangle + \varepsilon^2 \lambda^* \|\mathbf{v}\|^2$$

$$\text{whence} \quad \langle \mathbf{v}, \mathbf{X}(\mathbf{v}^* - \lambda^* \mathbf{v}^*) \rangle \leq \frac{\varepsilon}{2}(\|\mathbf{v}\|^2 - \langle \mathbf{v}, \mathbf{X}\mathbf{v} \rangle). \tag{5}$$

With $\boldsymbol{\mu}^* \equiv -\mathbf{X}\mathbf{v}^* + \lambda^* \mathbf{v}^*$, and taking $\varepsilon \to 0$ we obtain:

$$\langle \mathbf{v}, \boldsymbol{\mu}^* \rangle \geq 0 \quad \text{for all} \quad \mathbf{v} \in \mathbf{relint}(\mathcal{C}_{n,\mathbf{v}^*})$$

Since $\mathcal{C}_n$ is closed, so is $\mathcal{C}_{n,\mathbf{v}^*}$ and hence the above holds for every $\mathbf{v} \in \mathcal{C}_{n,\mathbf{v}^*}$. This implies that $\boldsymbol{\mu}^* \in \mathcal{C}_{n,\mathbf{v}^*}^*$. To prove Eq. (3) use $\mathbf{v} \in \mathcal{C}_{n,\mathbf{v}^*} \cap \operatorname{span}(\boldsymbol{\mu}^*)^\perp$ in Eq. (5).

Before proving the converse, we consider first the case when $\mathcal{C}_n$ is convex. By convexity of $\mathcal{C}_n$, we have that $\mathcal{C}_n \subseteq \mathcal{C}_{n,\mathbf{v}^*}$. Further since $\mathbf{v}^* \in \mathcal{C}_n$, we have that $\mathsf{P}_{\mathcal{C}_{n,\mathbf{v}^*}}(\mathbf{X}\mathbf{v}^*) = \mathsf{P}_{\mathcal{C}_n}(\mathbf{X}\mathbf{v}^*)$. Also, Remark 1.3 implies $\langle \boldsymbol{\mu}^*, \mathbf{v}^* \rangle = 0$. Together, these imply the Moreau decomposition claim.

In order to prove the converse, let $\mathbf{v} \in \mathcal{C}_n \cap \mathbb{S}^{n-1}$ and note that –as a consequence $\mathbf{w} \equiv \mathbf{v} - \mathbf{v}^* \in \mathcal{C}_{n,\mathbf{v}^*}$ with $2\langle \mathbf{w}, \mathbf{v}* \rangle = -\|\mathbf{w}\|_2^2$. We then have

$$\begin{aligned}
\langle \mathbf{v}, \mathbf{X}\mathbf{v} \rangle - \langle \mathbf{v}^*, \mathbf{X}\,\mathbf{v}^* \rangle &= 2\langle \mathbf{w}, \mathbf{X}\mathbf{v}^* \rangle + \langle \mathbf{w}, \mathbf{X}\,\mathbf{w} \rangle \\
&= 2\langle \mathbf{w}, \lambda^* \mathbf{v}^* - \boldsymbol{\mu}^* \rangle + \langle \mathbf{w}, \mathbf{X}\,\mathbf{w} \rangle \\
&= -2\langle \boldsymbol{\mu}^*, \mathbf{w} \rangle + \langle \mathbf{w}, (A - \lambda^* \mathrm{I})\,\mathbf{w} \rangle.
\end{aligned} \tag{6}$$

Letting $\mathbf{M} = \mathbf{X} - \lambda^* \mathrm{I}$, and $V \equiv \operatorname{span}(\boldsymbol{\mu}^*)^\perp$, we know by assumption that $\langle \mathbf{w}, \mathbf{M}\mathbf{w} \rangle \leq -2\Delta \, \|\mathbf{w}\|_2^2$ for all $\mathbf{w} \in \mathcal{D} \equiv \mathcal{C}_{n,\mathbf{v}^*} \cap V$, for some $\Delta > 0$. For a general $\mathbf{w} \in \mathcal{C}_{n,\mathbf{v}^*}$, let $\mathbf{w_0} \equiv \mathsf{P}_{\mathcal{D}}(\mathbf{w})$ and $\widetilde{\mathbf{v}} \equiv \mathbf{w} - \mathbf{w_0}$. We then have

$$\begin{aligned}
\langle \mathbf{w}, \mathbf{M}\mathbf{w} \rangle &= \langle \mathbf{w_0}, \mathbf{M}\mathbf{w_0} \rangle + 2\langle \widetilde{\mathbf{v}}, \mathbf{M}\mathbf{w_0} \rangle + \langle \widetilde{\mathbf{v}}, \mathbf{M}\widetilde{\mathbf{v}} \rangle \\
&\leq -2\Delta \|\mathbf{w_0}\|_2^2 + 2\|\mathbf{M}\|_2 \|\widetilde{\mathbf{v}}\|_2 \|\mathbf{w_0}\|_2 + \|M\|_2 \|\widetilde{\mathbf{v}}\|_2^2 \\
&\leq -2\Delta \|\mathbf{w_0}\|^2 + \gamma \|\mathbf{M}\|_2 \|\mathbf{w_0}\|_2^2 + \left(1 + \gamma^{-1}\right) \|\mathbf{M}\|_2 \|\widetilde{\mathbf{v}}\|_2^2,
\end{aligned}$$

where the last inequality follows for any $\gamma > 0$ by $2ab \leq \gamma\,a^2 + b^2/\gamma$. Setting $\gamma = \Delta/\|\mathbf{M}\|_2$ and $c = \|\mathbf{M}\|_2(1 + \gamma^{-1})$, we get

$$\langle \mathbf{w}, \mathbf{M}\mathbf{w} \rangle \leq -\Delta \|\mathbf{w_0}\|_2 + c\|\widetilde{\mathbf{v}}\|_2^2. \tag{7}$$

Now we claim that, for any $\delta > 0$ there exists $L = L(\mathcal{C}_{n,\mathbf{v}^*}, \boldsymbol{\mu}^*, \delta) \geq 0$ such that, for all $\mathbf{w} \in \mathcal{C}_{n,\mathbf{v}^*}$,

$$\|\widetilde{\mathbf{v}}\|_2^2 \leq \delta \|\mathbf{w_0}\|_2^2 + L\langle \boldsymbol{\mu}^*, \mathbf{w} \rangle^2. \tag{8}$$

Before proving this claim, let us show that it indeed implies the desired thesis. Setting $\delta = \Delta/c$, and $\widetilde{L} = c\,L(\mathcal{C}_{n,\mathbf{v}^*}, \boldsymbol{\mu}^*, \Delta/c)$, we conclude from Eq. (7) that

$$\langle \mathbf{w}, \mathbf{M}\mathbf{w} \rangle \leq \widetilde{L}\langle \boldsymbol{\mu}^*, \mathbf{w} \rangle^2.$$

Substituting this estimate in Eq. (6), we get

$$\langle \mathbf{v}, \mathbf{X}\,\mathbf{v} \rangle - \langle \mathbf{v}^*, \mathbf{X}\,\mathbf{v}^* \rangle \leq -2\langle \boldsymbol{\mu}^*, \mathbf{w} \rangle + \widetilde{L}\langle \boldsymbol{\mu}^*, \mathbf{w} \rangle^2 \,. \tag{9}$$

Hence, for all $\mathbf{v}$ such that $\|\mathbf{v} - \mathbf{v}^*\|_2 = \|\mathbf{w}\|_2 \leq 1/(\|\boldsymbol{\mu}^*\|_2 \widetilde{L})$ we have $0 \leq \langle \boldsymbol{\mu}^*, \mathbf{w} \rangle \leq 1/\widetilde{L}$ and therefore

$$\langle \mathbf{v}, \mathbf{X}\,\mathbf{v} \rangle - \langle \mathbf{v}^*, \mathbf{X}\,\mathbf{v}^* \rangle \leq -\langle \boldsymbol{\mu}^*, \mathbf{w} \rangle \leq 0 \,,$$

which completes our proof that $\mathbf{v}^*$ is a local maximum.

We are left with the task of proving the claim (8). Notice that, by scaling both sides, it is sufficient to prove it under the additional assumption $\|\mathbf{w}\|_2 = 1$. Fix $\delta > 0$ and assume by contradiction that the claim is false. Then, for each $k \in \mathbb{N}$ there exists $\mathbf{w}^{(k)} \in \mathcal{C}_{n,\mathbf{v}^*}$, $\|\mathbf{w}^{(k)}\|_2 = 1$ such that, letting $\mathbf{w_0}^{(k)} \equiv \mathsf{P}_{\mathcal{D}}(\mathbf{w}^{(k)})$ and $\widetilde{\mathbf{v}}^{(k)} \equiv \mathbf{w}^{(k)} - \mathbf{w_0}^{(k)}$ we get

$$\|\widetilde{\mathbf{v}}^{(k)}\|_2^2 > \delta \|\mathbf{w_0}^{(k)}\|_2^2 + k\langle \boldsymbol{\mu}^*, \mathbf{w}^{(k)} \rangle^2 \,. \tag{10}$$

Since $\{\mathbf{w}^{(k)}\}$ is a subset of the compact set $\mathcal{C}_{n,\mathbf{v}^*} \cap \mathbb{S}^{n-1}$, we can assume (by passing to a convergent subsequence), that $\mathbf{w}^{(k)} \to \mathbf{w}^{(\infty)}$. Since $\mathcal{C}_{n,\mathbf{v}^*} \cap \mathbb{S}^{n-1}$ is closed, $\mathbf{w}^{(\infty)} \in \mathcal{C}_{n,\mathbf{v}^*} \cap \mathbb{S}^{n-1}$. Further $\langle \boldsymbol{\mu}^*, \mathbf{w}^{(\infty)} \rangle^2 \leq \lim_{k\to\infty} \|\widetilde{\mathbf{v}}^{(k)}\|_2^2/k \leq \lim_{k\to\infty} 4/k = 0$. Hence $\mathbf{w}^{(\infty)} \in \mathcal{D}$ i.e. $\mathbf{w_0}^{(\infty)} = \mathbf{w}^{(\infty)}$, $\widehat{\mathbf{v}}^{(\infty)} = 0$. Further taking the limit $k \to \infty$ in (10) we get

$$\|\widetilde{\mathbf{v}}^{(\infty)}\|_2^2 \geq \delta \|\mathbf{w_0}^{(\infty)}\|_2^2$$

i.e. $0 \geq \delta$. Since we chose $\delta > 0$ this gives the desired contradiction. $\qquad\square$

The above proof implies a quantitative bound on the radius of the neighborhood within which $\mathbf{v}^*$ is an optimum.

**Corollary 1.5.** *Assume the conditions*

$$\mathbf{X}\mathbf{v}^* = \lambda^* \, \mathbf{v}^* - \boldsymbol{\mu}^* \,,$$

$$\lambda^* - 2\Delta \geq \sup \left\{ \langle \mathbf{v}, \mathbf{X}\mathbf{v} \rangle : \mathbf{v} \in \mathcal{C}_{n,\mathbf{v}^*} \cap \operatorname{span}(\boldsymbol{\mu}^*)^\perp, \|\mathbf{v}\|_2 = 1 \right\},$$

*hold, and further assume that $L(\mathcal{C}_{n,\mathbf{v}^*}, \boldsymbol{\mu}^*, \delta) \geq 0$ is such that, for all $\mathbf{w} \in \mathcal{C}_{n,\mathbf{v}^*}$,*

$$\|\widetilde{\mathbf{v}}\|_2^2 \leq \delta \|\mathbf{w_0}\|_2^2 + L\langle \boldsymbol{\mu}^*, \mathbf{w} \rangle^2 \,.$$

*Let $\mathbf{M} = \mathbf{X} - \lambda_* \mathrm{I}$, $c = \Delta \|\mathbf{M}\|_2 (\Delta + \|\mathbf{M}\|_2)$, and $\widetilde{L} = c\, L(\mathcal{C}_{n,\mathbf{v}^*}, \boldsymbol{\mu}^*, \Delta/c)$. Then $\mathbf{v}^*$ is a global maximum of the optimization problem (1) within a neighborhood $\mathsf{Ball}(\mathbf{v}^*, 1/(\widetilde{L}\|\boldsymbol{\mu}^*\|_2))$.*

## 2 Proof of Theorem 1

Recall the model assumptions for the data $\mathbf{X}$:

$$\mathbf{X} = \beta \mathbf{v_0} \mathbf{v_0}^\mathsf{T} + \mathbf{Z}.$$

Here we assume that $\mathbf{v_0} \in \mathcal{C}_n \cap \mathbb{S}^{n-1}$ and $\mathbf{Z}_{ij} \sim \mathsf{N}(0, 1/n)$ are independent, up to symmetry. We first define the following useful quantities:

**Definition 2.1.** *For a cone $\mathcal{C}_n \subseteq \mathbb{R}^n$ we define its normalized Gaussian width as:*

$$\omega(\mathcal{C}_n) \equiv \frac{1}{\sqrt{n}} \mathbb{E}\{ \sup_{\mathbf{v} \in \mathcal{C}_n \cap \mathbb{S}^{n-1}} \langle \mathbf{g}, \mathbf{v} \rangle \},$$

*where $\mathbf{g} \sim \mathsf{N}(0, \mathrm{I}_n)$.*

**Definition 2.2.** *For a cone $\mathcal{C}_n \subseteq \mathbb{R}^n$, we define its packing number $N(\mathcal{C}_n, \varepsilon)$ as the size of the maximal subset $X$ of $\mathcal{C}_n \cap \mathbb{S}^{n-1}$ such that for every $\mathbf{v}_1, \mathbf{v}_2 \in \mathcal{C}_n \cap \mathbb{S}^{n-1}$, $\|\mathbf{v}_1 - \mathbf{v}_2\| > \varepsilon$.*

We have the following useful facts:

**Lemma 2.3.** *There exist universal constants $c_1, c_2$ such that:*

$$c_1 \sup_\varepsilon \varepsilon \sqrt{\log N(\mathcal{C}_n, \varepsilon)} \leq \sqrt{n} \omega(\mathcal{C}_n) \leq c_2 \inf_{\varepsilon \leq 1} \{2\sqrt{n}\varepsilon(1 + \sqrt{\log(2/\varepsilon)}) + (2 - \varepsilon)\sqrt{\log N(\mathcal{C}_n, \varepsilon)}\}.$$

*Proof.* The left hand inequality is the Sudakov minoration inequality. For the latter, we first employ Dudley inequality:

$$\sqrt{n}\omega(\mathcal{C}_n) \leq c_3 \int_0^\infty \sqrt{\log N(\mathcal{C}_n, \varepsilon)} \mathrm{d}\varepsilon,$$

for a universal constant $c_3$. We know that the diameter of $\mathbb{S}^{n-1}$ is 2. Further as $\mathcal{C}_n \cap \mathbb{S}^{n-1} \subseteq \mathbb{S}^{n-1}$ using a standard volume packing argument for $\mathbb{S}^{n-1}$ [] we have that:

$$N(\mathcal{C}_n, \varepsilon) \leq \left(1 + \frac{2}{\varepsilon}\right)^n.$$

Thus for any $0 \leq \varepsilon \leq 1$:

$$\sqrt{n}\omega(\mathcal{C}_n) \leq c_3 \left[\int_0^\varepsilon \sqrt{n \log\left(1 + \frac{2}{u}\right)} \mathrm{d}u + \int_\varepsilon^2 \sqrt{N(\mathcal{C}_n, \varepsilon)} \mathrm{d}u\right].$$

We simplify the first integral:

$$\int_0^\varepsilon \sqrt{\log\left(1 + \frac{2}{u}\right)} \mathrm{d}u \leq 2 \int_0^\varepsilon \sqrt{\log\left(\frac{2}{u}\right)} \mathrm{d}u$$

$$= 2 \int_{\sqrt{\log(2/\varepsilon)}}^\infty (4y^2) \exp(-y^2) \mathrm{d}y$$

$$\leq 2\varepsilon \left(1 + \sqrt{\log\left(\frac{2}{\varepsilon}\right)}\right),$$

using standard integration by parts. Since $N(\mathcal{C}, \varepsilon)$ is monotone nonincreasing in $\varepsilon$, we have that the second integral is bound above by $(2 - \varepsilon)N(\mathcal{C}_n, \varepsilon)$. These estimates imply the thesis using the observation that $\varepsilon$ is arbitrarily chosen.

$\square$

The following lemma is proved in [ALMT13].

**Lemma 2.4.** *For any closed convex cone:*

$$\omega(\mathcal{C}_n)^2 \leq \delta(\mathcal{C}_n) \leq \omega(\mathcal{C}_n)^2 + \frac{1}{n}.$$

We can now prove Theorem 1

*Proof of Theorem 1.* Let $X$ denote a maximal $\varepsilon$-net of $\mathcal{C}_n \cap \mathbb{S}^{n-1}$ as in Lemma 2.3, for $\varepsilon$ to be fixed later in the proof. Hence, for any $\mathbf{v}_i, \mathbf{v}_j \in X$ distinct, we have $\|\mathbf{v}_i - \mathbf{v}_j\| > \varepsilon$ or, equivalently, $\langle \mathbf{v}_i, \mathbf{v}_j \rangle < 1 - \varepsilon^2/2$. Note that the maximality of $X$ implies also that it is an $\varepsilon$-cover of $\mathcal{C}_n \cap \mathbb{S}^{n-1}$, i.e. for any $\mathbf{v} \in \mathcal{C}_n \cap \mathbb{S}^{n-1} \min_{\mathbf{v}' \in X} \|\mathbf{v} - \mathbf{v}'\| \leq \varepsilon$.

Let $\mathbf{v}_0$ be uniformly distributed in the set $X$. For an estimator $\widehat{\mathbf{v}}(\mathbf{X}) \in \mathcal{V}$, we define $G(\widehat{\mathbf{v}}(Y)) = \arg\min_{\mathbf{v} \in X} \|\widehat{\mathbf{v}}(\mathbf{X}) - \mathbf{v}\|_2$. We now bound the probability of the error event $\{G(\widehat{\mathbf{v}}(\mathbf{X})) \neq \mathbf{v}_0\}$. By definition of $G(\widehat{\mathbf{v}}(\mathbf{X}))$, the event $G(\widehat{\mathbf{v}}(Y)) \neq \mathbf{v}_0$ implies that there exists $\mathbf{v}_i \in X, \mathbf{v}_i \neq \mathbf{v}_0$ such that $\|\widehat{\mathbf{v}}(\mathbf{X}) - \mathbf{v}_i\|_2 \leq \|\widehat{\mathbf{v}}(\mathbf{X}) - \mathbf{v}_0\|_2$. This along with triangle inequality implies that $\varepsilon < \|\mathbf{v}_i - \mathbf{v}_0\| \leq 2\|\widehat{\mathbf{v}}(\mathbf{X}) - \mathbf{v}_0\|$, i.e. $\|\widehat{\mathbf{v}}(\mathbf{X}) - \mathbf{v}_0\|_2 > \varepsilon/2$. By Markov inequality we have:

$$\mathbb{P}\{G(\widehat{\mathbf{v}}(\mathbf{X})) \neq \mathbf{v}_0\} \leq \mathbb{P}\{\|\widehat{\mathbf{v}}(\mathbf{X}) - \mathbf{v}_0\|_2 > \varepsilon/2\}$$

$$\leq 4 \frac{\mathbb{E}\{\|\widehat{\mathbf{v}}(\mathbf{X}) - \mathbf{v}_0\|_2^2\}}{\varepsilon^2}$$

$$= \frac{8(1 - \mathbb{E}\{\langle \widehat{\mathbf{v}}(\mathbf{X}), \mathbf{v}_0 \rangle\})}{\varepsilon^2}$$

$$\leq \frac{8\mathsf{R}(\widehat{\mathbf{v}}(\mathbf{X}); \mathbf{v}_0)}{\varepsilon^2}, \tag{11}$$

By Fano's inequality we have that:

$$\mathbb{P}\{G(\widehat{\mathbf{v}}(\mathbf{X}) \neq \mathbf{v_0}\} \geq 1 - \frac{\gamma + \log 2}{\log |X|},$$

where $\gamma = \max_{x \neq x'} D(P_{\mathbf{v}} \| P_{\mathbf{v}'})$ where $D(\cdot \| \cdot)$ is the Kullback-Liebler divergence and $P_{\mathbf{v}}$ denotes the law of $\mathbf{X}$ conditional on $\mathbf{v_0} = \mathbf{v}$. Conditional on $\mathbf{v_0} = \mathbf{v}$, $\mathbf{X}$ has mean $\mathbf{v}\mathbf{v}^\mathsf{T}$ and has Gaussian entries with variance $1/n$. A standard calculation implies that:

$$\begin{aligned} D(P_{\mathbf{v}} \| P_{\mathbf{v}'}) &\leq n\beta^2 \|\mathbf{v}\mathbf{v}^\mathsf{T} - \mathbf{v}'\mathbf{v}'^\mathsf{T}\|_F^2 \\ &= 2n\beta^2(1 - \langle \mathbf{v}, \mathbf{v}' \rangle^2) \\ &\leq 2n\beta^2. \end{aligned}$$

We have using Lemma 2.3 that $\log |X| \geq cn(\omega(\mathcal{C}_n) - 2\varepsilon(1 + \sqrt{\log(2/\varepsilon)}))^2$. Combining this with Fano's inequality above and Eq.(11):

$$\mathsf{R}(\widehat{\mathbf{v}}(\mathbf{X}); \mathbf{v_0}) \geq \frac{\varepsilon^2}{8}\left(1 - \frac{2\beta^2}{c(\omega(\mathcal{C}_n) - 2\varepsilon(1 + \sqrt{\log(2/\varepsilon)}))^2}\right).$$

Further, the minimax risk satisfies $\mathsf{R}(\mathcal{C}_n) \geq \mathsf{R}(\widehat{\mathbf{v}}(\mathbf{X}); \mathbf{v_0}))$ for some estimator $\widehat{\mathbf{v}}(\mathbf{X})$ and the above bound is uniformly true for all estimators. Using $\varepsilon_* = 1/4\omega(\mathcal{C}_n)/\sqrt{\log(1/\omega(\mathcal{C}_n))}$ implies that $2\varepsilon(1 + \sqrt{\log(2/\varepsilon)}) \leq \omega(\mathcal{C}_n)/2$. Hence:

$$\mathsf{R}(\mathcal{C}_n) \geq c'\frac{\omega(\mathcal{C}_n)^2}{\log(1/\omega(\mathcal{C}_n))},$$

when $\beta \leq c''\omega(\mathcal{C}_n)$ for some universal constants $c'$, $c''$. Applying Lemma 2.4 and the constraint $\omega(\mathcal{C}_n) \geq \sqrt{2/n}$ implies the result for appropriately adjusted $c'$, $c''$.

$\square$

## 3   Proof of Theorem 2

We first prove the following useful lemma based on standard Gaussian comparisons.

**Lemma 3.1.** *Assume that the noise $\mathbf{Z}$ has i.i.d. $\mathsf{N}(0, 1/n)$ entries up to symmetry. Then, with probability at least $1 - n^{-5}$*

$$\lambda_1(Z; \mathcal{C}_n) \leq 2\sqrt{\delta(\mathcal{C}_n)} + \sqrt{\frac{10 \log n}{n}}. \tag{12}$$

*Proof.* We will apply the Sudakov-Fernique inequality (see e.g. [Vit00, Theorem 1] ) to the two processes $\{\mathbf{M}(\mathbf{v})\}$, $\{\mathbf{V}(\mathbf{v})\}$ indexed by $x \in \mathcal{C}_n \cap \mathbb{S}^{n-1}$ defined as follows:

$$\mathbf{M}(\mathbf{v}) \equiv \langle \mathbf{v}, \mathbf{Z}\mathbf{v} \rangle \quad \text{and} \quad \mathbf{V}(\mathbf{v}) \equiv 2\langle \mathbf{v}, \mathbf{g} \rangle,$$

for a random vector $\mathbf{g} \sim \mathsf{N}(0, \mathbf{I}_n/n)$ and $\mathbf{Z}$ being a standard normal matrix. Basic algebra gives for all $\mathbf{v}$, $\mathbb{E}\mathbf{M}(\mathbf{v}) = \mathbb{E}\mathbf{V}(\mathbf{v}) = 0$ and

$$\mathbb{E}\{[\mathbf{M}(\mathbf{v}_1) - \mathbf{M}(\mathbf{v}_2)]^2\} = \frac{4}{n}(1 - \langle \mathbf{v}_1, \mathbf{v}_2 \rangle^2), \quad \mathbb{E}\{[\mathbf{V}(\mathbf{v}_1) - \mathbf{V}(\mathbf{v}_2)]^2\} = \frac{8}{n}(1 - \langle \mathbf{v}_1, \mathbf{v}_2 \rangle).$$

Hence, by using the fact that for $a \in [-1, 1]$, the inequality $1 - a^2 \leq 2(1 - a)$ holds, we get $\mathbb{E}\{[\mathbf{M}(\mathbf{v}_1) - \mathbf{M}(\mathbf{v}_2)]^2\} \leq \mathbb{E}\{[\mathbf{V}(\mathbf{v}_1) - \mathbf{V}(\mathbf{v}_2)]^2\}$. We conclude using Sudakov-Fernique comparison lemma that:

$$\mathbb{E}\max\{\langle \mathbf{v}, \mathbf{Z}\mathbf{v} \rangle : \mathbf{v} \in \mathcal{C}_n \cap \mathbb{S}^{n-1}\} \leq 2\,\mathbb{E}\max\{\langle \mathbf{v}, \mathbf{g} \rangle : \mathbf{v} \in \mathcal{C}_n \cap \mathbb{S}^{n-1}\}$$

By using Proposition 10.1 from [ALMT13], $\mathbb{E}\max\{\langle \mathbf{v}, \mathbf{g} \rangle : \mathbf{v} \in \mathcal{C}_n \cap \mathbb{S}^{n-1}\} \leq \sqrt{\delta(\mathcal{C}_n)}$, we get that

$$\mathbb{E}\lambda_1(\mathbf{Z}; \mathcal{C}_n) \leq 2\sqrt{\delta(\mathcal{C}_n)}. \tag{13}$$

By using the fact that $\mathbf{Z} \mapsto \max\left\{\langle \mathbf{v}, \mathbf{Z}\mathbf{v}\rangle \ : \ \mathbf{v} \in \mathcal{C}_n \cap \mathbb{S}^{n-1}\right\}$ is 1-Lipschitz and concentration inequality on $\max\left\{\langle \mathbf{v}, \mathbf{Z}\mathbf{v}\rangle \ : \ \mathbf{v} \in \mathcal{C}_n \cap \mathbb{S}^{n-1}\right\}$, with probability at least $1 - \exp\{-t^2 n/2\}$, we have

$$\lambda_1(\mathbf{Z}; \mathcal{C}_n) \leq 2\sqrt{\delta(\mathcal{C}_n)} + t \ .$$

Take $t = \sqrt{(10 \log n)/n}$ and the claim follows.

$\square$

We can now prove Theorem 2

*Proof.* By optimality of $\widehat{\mathbf{v}}^{\mathrm{ML}}$, we have

$$\langle \widehat{\mathbf{v}}^{\mathrm{ML}}, \mathbf{X}\widehat{\mathbf{v}}^{\mathrm{ML}}\rangle \geq \langle \mathbf{v_0}, \mathbf{X}\mathbf{v_0}\rangle = \beta + \langle \mathbf{v_0}, \mathbf{Z}\mathbf{v_0}\rangle$$
$$\geq \beta - \lambda_1(\mathbf{Z}, \mathcal{C}_n).$$

On the other hand

$$\langle \widehat{\mathbf{v}}^{\mathrm{ML}}, \mathbf{X}\widehat{\mathbf{v}}^{\mathrm{ML}}\rangle = \beta\langle \mathbf{v_0}, \widehat{\mathbf{v}}^{\mathrm{ML}}\rangle^2 + \langle \widehat{\mathbf{v}}^{\mathrm{ML}}, \mathbf{Z}\widehat{\mathbf{v}}^{\mathrm{ML}}\rangle \leq \beta\langle \mathbf{v_0}, \widehat{\mathbf{v}}^{\mathrm{ML}}\rangle^2 + \lambda_{\max}(\mathbf{Z}; \mathcal{C}_n) \ .$$

The claim that $\mathsf{R}_{\mathcal{C}_n}(\widehat{\mathbf{v}}^{\mathrm{ML}}; \mathcal{C}_n) \leq 4(\sqrt{\delta(\mathcal{C}_n)} + \varepsilon_n)/\beta$ follows simply by putting together the above inequalities along with the previous lemma for $\lambda_{\max}(\mathbf{Z}; \mathcal{C}_n)$.

For the other claim we have as above:

$$\langle \widehat{\mathbf{v}}^{\mathrm{ML}}, \mathbf{X}\widehat{\mathbf{v}}^{\mathrm{ML}}\rangle \geq \beta + \langle \mathbf{v_0}, \mathbf{Z}\mathbf{v_0}\rangle,$$
$$\langle \widehat{\mathbf{v}}^{\mathrm{ML}}, \mathbf{X}\widehat{\mathbf{v}}^{\mathrm{ML}}\rangle = \beta\langle \mathbf{v_0}, \widehat{\mathbf{v}}^{\mathrm{ML}}\rangle^2 + \langle \widehat{\mathbf{v}}^{\mathrm{ML}}, \mathbf{Z}\widehat{\mathbf{v}}^{\mathrm{ML}}\rangle.$$

Together, this implies:

$$\beta(1 - \langle \mathbf{v_0}, \widehat{\mathbf{v}}^{\mathrm{ML}}\rangle^2) \leq \langle \mathbf{Z}, -\mathbf{v_0}\mathbf{v_0}^{\mathsf{T}} + \widehat{\mathbf{v}}^{\mathrm{ML}}(\widehat{\mathbf{v}}^{\mathrm{ML}})^{\mathsf{T}}\rangle$$
$$\leq \|\mathbf{Z}\|_2\|\mathbf{v_0}\mathbf{v_0}^{\mathsf{T}} - \widehat{\mathbf{v}}^{\mathrm{ML}}(\widehat{\mathbf{v}}^{\mathrm{ML}})^{\mathsf{T}}\|_*,$$

where the last line follows from Holder's inequality and $\|\cdot\|_*$ denotes the nuclear norm (or sum of singular values). Since $\mathbf{v_0}\mathbf{v_0}^{\mathsf{T}} - \widehat{\mathbf{v}}^{\mathrm{ML}}(\widehat{\mathbf{v}}^{\mathrm{ML}})^{\mathsf{T}}$ is has rank at most two:

$$\|\mathbf{v_0}\mathbf{v_0}^{\mathsf{T}} - \widehat{\mathbf{v}}^{\mathrm{ML}}(\widehat{\mathbf{v}}^{\mathrm{ML}})^{\mathsf{T}}\|_* \leq \sqrt{2}\|\mathbf{v_0}\mathbf{v_0}^{\mathsf{T}} - \widehat{\mathbf{v}}^{\mathrm{ML}}(\widehat{\mathbf{v}}^{\mathrm{ML}})^{\mathsf{T}}\|_F$$
$$= 2\sqrt{1 - \langle \mathbf{v_0}, \widehat{\mathbf{v}}^{\mathrm{ML}}\rangle^2}.$$

Thus we have:

$$1 - \langle \mathbf{v_0}, \widehat{\mathbf{v}}^{\mathrm{ML}}\rangle^2 \leq \frac{2\|\mathbf{Z}\|^2}{\beta^2}.$$

Using the fact that $1 - a^2 \geq 1 - |a|$ when $a \in [-1, 1]$, we obtain the desired result for the risk $\mathsf{R}(\widehat{\mathbf{v}}^{\mathrm{ML}}; \mathcal{C}_n)$.

$\square$

## 4 Proof of Theorem 3

Note that the problem (1) is unchanged (except for an additive constant in the objective function) if we replace $\mathbf{X}$ with $\mathbf{X}_\rho = \mathbf{X} + \rho\,\mathbf{I}$. We will take advantage of this freedom and consider the modified iteration

$$\widehat{\mathbf{v}}^{t+1} = \frac{\mathsf{P}_{\mathcal{C}}(\mathbf{u}^t)}{\|\mathsf{P}_{\mathcal{C}}(\mathbf{u}^t)\|_2}, \tag{14}$$
$$\mathbf{u}^t = \mathbf{X}_\rho\widehat{\mathbf{v}}^t . \tag{15}$$

It is convenient to define

$$\boldsymbol{\mu}^{t+1} \equiv -\mathsf{P}_{\mathcal{C}_n^\circ}(\mathbf{X}_\rho\widehat{\mathbf{v}}^t), \tag{16}$$
$$\lambda_{t+1} \equiv \left\|\mathsf{P}_{\mathcal{C}_n}(\mathbf{X}_\rho\widehat{\mathbf{v}}^t)\right\|_2 . \tag{17}$$

Then we have the identity

$$\mathbf{X}_\rho \widehat{\mathbf{v}}^{t-1} = \lambda_t \widehat{\mathbf{v}}^t - \boldsymbol{\mu}^t, \tag{18}$$

which is the Moreau's decomposition of $\mathbf{X}_\rho \widehat{\mathbf{v}}^{t-1}$. In particular

$$\langle \widehat{\mathbf{v}}^t, \boldsymbol{\mu}^t \rangle = 0. \tag{19}$$

**Lemma 4.1.** *With the above definitions we have*

$$\lambda_t = \lambda_{t+1} \langle \widehat{\mathbf{v}}^{t+1}, \widehat{\mathbf{v}}^{t-1} \rangle - \langle \boldsymbol{\mu}^{t+1}, \widehat{\mathbf{v}}^{t-1} \rangle \tag{20}$$

$$\leq \lambda_{t+1} \langle \widehat{\mathbf{v}}^{t+1}, \widehat{\mathbf{v}}^{t-1} \rangle. \tag{21}$$

*Proof.* Taking the scalar product of both sides of Eq. (18) by $\widehat{\mathbf{v}}^t$ and using Eq. (19), together with the fact that $\|\widehat{\mathbf{v}}^t\|_2 = 1$ by construction, we get

$$\lambda_t = \langle \widehat{\mathbf{v}}^t, \mathbf{X} \widehat{\mathbf{v}}^{t-1} \rangle = \langle \widehat{\mathbf{v}}^{t-1}, \mathbf{X} \widehat{\mathbf{v}}^t \rangle \tag{22}$$

$$= \langle \widehat{\mathbf{v}}^{t-1}, \lambda_{t+1} \widehat{\mathbf{v}}^{t+1} - \mu^{t+1} \rangle. \tag{23}$$

This proves Eq. (20). Equation (21) follows from $\widehat{\mathbf{v}}^{t-1} \in \mathcal{C}_n$, $\boldsymbol{\mu}^{t+1} \in \mathcal{C}_n^*$, which imply $\langle \widehat{\mathbf{v}}^{t-1}, \mu^{t+1} \rangle \geq 0$. ☐

**Lemma 4.2.** *For any $t \geq 1$, we have*

$$\rho + \lambda_{\min}(\mathbf{X}) \leq \lambda_t \leq \rho + \lambda_{\max}(\mathbf{X}).$$

*Proof.* For $\mathbf{y} \in \mathcal{C}_n^\circ$, $\max\{\langle \mathbf{y}, \mathbf{z} \rangle : \mathbf{z} \in \mathcal{C}_n \cap \mathbb{S}^{n-1}\} = 0$ and $\mathsf{P}_{\mathcal{C}_n}(\mathbf{y}) = 0$. For any $\mathbf{y} \notin \mathcal{C}_n^\circ$, $\mathsf{P}_{\mathcal{C}_n}(\mathbf{y}) \neq 0$, so by using Cauchy-Schwarz inequality, and the fact that $\mathsf{P}_{\mathcal{C}_n}(\mathbf{y})/\|\mathsf{P}_{\mathcal{C}_n}(\mathbf{y})\|_2 \in \mathcal{C}_n \cap \mathbb{S}^{n-1}$,

$$\|\mathsf{P}_{\mathcal{C}_n}(\mathbf{y})\|_2 = \max\{\langle \mathbf{y}, \mathbf{z} \rangle : \mathbf{z} \in \mathcal{C}_n \cap \mathbb{S}^{n-1}\}.$$

Then, since $\widehat{\mathbf{v}}^{t-1} \in \mathcal{C}_n \cap \mathbb{S}^{n-1}$, we have

$$\lambda_t = \|\mathsf{P}_{\mathcal{C}_n}(\mathbf{X}_\rho \widehat{\mathbf{v}}^{t-1})\|_2 \geq \langle \widehat{\mathbf{v}}^{t-1}, \mathbf{X}_\rho \widehat{\mathbf{v}}^{t-1} \rangle,$$

which yields the desired lower bound by definition. The upper bound follows since $\lambda_t \leq \|\mathbf{X}_\rho \widehat{\mathbf{v}}^{t-1}\|_2$. ☐

**Proposition 4.3.** *Assume $\rho + \lambda_{\min}(\mathbf{X}) > 0$. Then the sequence $\{\lambda_t\}_{t \geq 0}$ is bounded an non-decreasing and therefore it has a limit*

$$\lambda_* = \lim_{t \to \infty} \lambda_t = \lim_{t \to \infty} \langle \widehat{\mathbf{v}}^t, \mathbf{X}_\rho \widehat{\mathbf{v}}^t \rangle. \tag{24}$$

*Further, if $\widehat{\mathbf{v}}_*$ is any sub-sequential limit of $\{\widehat{\mathbf{v}}^t\}_{t \geq 0}$ (i.e. $\lim_{k \to \infty} \widehat{\mathbf{v}}^{t(k)} = \widehat{\mathbf{v}}_*$ for some sequence $\{t(k)\}$) then it satisfies the stationarity condition*

$$\mathbf{X} \widehat{\mathbf{v}}_* = (\lambda_* - \rho) \widehat{\mathbf{v}}_* - \boldsymbol{\mu}_*. \tag{25}$$

*Proof.* The existence of the limit $\lambda_*$ follows immediately from Lemma 4.1 and 4.2. Next multiplying the identity (18) by $\widehat{\mathbf{v}}^t$, we get, using Cauchy-Schwartz,

$$\lambda_t = \langle \widehat{\mathbf{v}}^t, \mathbf{X}_\rho \widehat{\mathbf{v}}^{t-1} \rangle \leq \sqrt{\langle \widehat{\mathbf{v}}^t, \mathbf{X}_\rho \widehat{\mathbf{v}}^t \rangle \langle \widehat{\mathbf{v}}^{t-1}, \mathbf{X}_\rho \widehat{\mathbf{v}}^{t-1} \rangle}.$$

Multiplying Eq. (18) by $\widehat{\mathbf{v}}^{t-1}$, we get $\langle \widehat{\mathbf{v}}^{t-1}, \mathbf{X} \widehat{\mathbf{v}}^{t-1} \rangle \leq \lambda_t \langle \widehat{\mathbf{v}}^t, \widehat{\mathbf{v}}^{t-1} \rangle$ and, changing the iteration number, $\langle \widehat{\mathbf{v}}^t, \mathbf{X} \widehat{\mathbf{v}}^t \rangle \leq \lambda_{t+1} \langle \widehat{\mathbf{v}}^t, \widehat{\mathbf{v}}^{t+1} \rangle$. Substituting in the above, we obtain

$$\lambda_t \leq \lambda_{t+1} \langle \widehat{\mathbf{v}}^t, \widehat{\mathbf{v}}^{t-1} \rangle \langle \widehat{\mathbf{v}}^t, \widehat{\mathbf{v}}^{t+1} \rangle.$$

Since $\lambda_t \to \lambda_*$, we conclude that $\langle \widehat{\mathbf{v}}^t, \widehat{\mathbf{v}}^{t-1} \rangle \to 1$ or

$$\lim_{t \to \infty} \|\widehat{\mathbf{v}}^t - \widehat{\mathbf{v}}^{t+1}\|_2 = 0.$$

Using this in the identity $\lambda_t = \langle \widehat{\mathbf{v}}^t, \mathbf{X}_\rho \widehat{\mathbf{v}}^{t-1} \rangle$ derived above, we deduce that $\langle \widehat{\mathbf{v}}^t, \mathbf{X}_\rho \widehat{\mathbf{v}}^t \rangle \to \lambda_*$. Using it in Eq. (18) (with $t = t(k)$ as per the statement) we get the stationarity condition (25). ☐

**Remark 4.4.** Corollary 4.1 follows directly from the proof of Theorem 3 and the Gaussian comparison lemma 3.1.