[Reviews · NeurIPS 2014]

Submitted by Assigned_Reviewer_32

This paper considers the estimation of an unknown vector v0 from noisy quadratic observations and some additional information regarding v0. Specifically, it considers that the unknown vector v0 is from a convex cone. It rigorously shows that the resulting optimization problem is tractable. Note that the resulting optimization problem in Eq.(3) is non-convex. But the authors approach this problem from the Gaussian dimension of the cone.
This paper is a very theoretical paper. Many important results are shown in the supplementary file, which is 28 pages long. The main paper actually is very brief. It simply motivates the work and shows the main results of this paper. I did not check every detail of the proof, but it sounds good to me.
Overall, this is a fairly interesting paper. It addresses an important problem and show it is tractable with rigorous proof. I think the quality of the paper is good enough for NIPS. The paper is brief, and all proofs are in the supplementary file, making it not easy to read it.
Summary: It is a good theoretical paper which shows that the estimation of an unknown vector v0 given noisy quadratic observations and the condition that v0 is in a convex cone. It shows an interesting and useful result although all proofs are in the supplementary file.

Submitted by Assigned_Reviewer_41

This paper studies the recovery of unknown vectors belonging to a convex cone from noisy quadratic observations. The contributions are three-fold, namely 1) establishing information-theoretic limits on the recovery of the unknown vectors, 2) assessing the quality of maximum likelihood estimates and 3) proposing a low-complexity iterative estimator.

The exposition is well presented and the mathematics look correct. Due to time constraints it was not possible to carefully check the Supplemental Material in all its extension.

My major concern is regarding the applicability and relevance of the presented results beyond mathematical beauty. I found the motivation of cone-constraining vague throughout the manuscript, and the experimental evaluation is solely carried out on a synthetic dataset designed to fulfil the author's assumptions.
Summary: Information-theoretic limits for the estimation of cone-constrained principal component analysis are provided. Pros: The exposition is well presented and the mathematics are beautiful. Cons: motivation/relevance is vague throughout the manuscript and experimental evaluation is borderline.

Submitted by Assigned_Reviewer_42

The authors consider the model of noisy quadratic measurements (rank1 matrix plus noise), where we want to estimate the spike that generates the rank-1 matrix. They refer to this model as a quadratic observation model.

Similar estimation problems that come with non-convex constraints (i.e., sparse PCA) have been analyzed where there is a gap between minimax risk and tractable estimators. Here the authors consider convex cone constraints, (which don't render the problem any easier at first sight). Although the ML-optimization problem is computationally intractable, in the general case, they can show that a projected version of the power method is almost optimal in terms of minimax risk estimation, and approximates the ML performance within a constant gap.

In the case of the nonnegative cone, they show that their analysis can be tightened to an almost zero gap between the ML and the projected power method.

In summary, their contribution is three fold:
- they present the information theoretic bounds for the estimation problem
- they present the risk of the ML estimator using the gaussian dimension of the involved cone
- they use a projected power method algorithm to essentially achieve ML performance.

I generally really liked the paper; it’s an original and significant contribution to the relevant literature. I think that the use of the gaussian dimension of the cone for quadratic measurement problems seems new and very promising. The authors bound the minimax rate using that precisely. I wonder if there is any relevance to the recent phase recovery problem.

Similarity with concurrent works of [1] and [2]:
Some of the results look very similar to some recent works [1] and [2], where information theoretic bounds and a similar approximate message passing/ projected power method was used. It seems that section 5 establishes the same result as the work of [1]. However, the submission date of [1] is later than the present manuscript was submitted, and the works can be considered concurrent.

The presented work here however is much more general and very significant to the relevant literature; the above comment is only made due to the relevance between the results.

Some typos:
typo in line 137, reference is missing
equation (3), subjectto -> subject to

[1] “Non-negative Principal Component Analysis: Message Passing Algorithms and Sharp Asymptotics”
Andrea Montanari, Emile Richard
http://arxiv.org/pdf/1406.4775v1.pdf

[2] “Information-theoretically Optimal Sparse PCA”
Yash Deshpande, Andrea Montanari
http://arxiv.org/pdf/1402.2238v2.pdf

Summary: This is a well written paper, lengthy and rich theoretical study, with very significant results.
Author Feedback
Author rebuttal: We thank the reviewer for their feedback of our paper.

Referee 2 raises the issue of the applicability of cone-constrained PCA.

Here is a partial list of applications that we have in mind and can be addressed as applications of our general framework:[Notice that a few applications emerging from earlier literature are already mentioned, see refs. [16] and [19-23] ]

1) In neural signal processing, PCA is routinely used to find a low-dimensional space onto which spike signals are projected (and subsequently clustered or classified). Physiological considerations impose several constraints on the `noiseless' spikes that PCA attempts to recover. For instance, a spike is first positive and then negative, and so on. Several of these constraints can be modeled as convex cone constraints: we carried out a preliminary analysis that indeed suggests improved estimation.

2) Similar constraints appear in other time series data. As a toy example, given temperatures recorded at various locations on the globe, one might want to test whether there is a global increasing trend. A naive approach would be to average all the time series, but this is obviously suboptimal: temperatures at locations with different climates are not directly comparable. PCA effectively performs a weighted average, with weights derived from the data. The increasing trend can be modeled as a convex-cone constraint (the set of increasing functions is a convex cone). Note that while the paper focuses on estimation, hypothesis testing is a straightforward extension.

3) In microarray experiments, gene expression levels are simultaneously measured for thousands of genes over two groups of patients. Similar data matrices are produced from DNA sequencing. It is common to search for a set of genes that are differentially expressed across the two groups, for instance that are expressed more than average in group 1, and less in group 2. While PCA is routinely used for this task, cone-constrained PCA can lead to greater statistical efficiency (the sign constraints are convex cones).

4) Finding low-rank approximations of tensors has multiple applications in machine learning, but is extremely challenging from the point of view of computational complexity. Here is an approach that builds on the present work, and that we are currently investigating. Consider --say-- the case of a rank 4 symmetric tensor A (it is easy to generalize to tensors of other orders). We want to maximize \sum_ijkl A_ijkl x_i x_j x_k x_l s.t. \|x\| =1. Here is a relaxation, that introduces new variables S_ij (relaxing x_ix_j): maximize \sum_ijkl A_ijkl S_ij S_kl s.t. \|S\|_F = 1, S\in PSD where PSD is the cone of positive semidefinite matrices. This is a non-convex relaxation! However is a convex-cone constrained PCA problem (in n*(n+1)/2 dimensions). Our results indicates that is possible to solve it efficiently.

5) The above list only concerns convex cones. An equally important set of applications can be modeled as non-convex cones (e.g. sparse PCA). Note first of all that our results on the maximum likelihood estimator cover this case as well. From the algorithmic point of view, convex cone optimization can be used in various ways in those problems, e.g. as relaxations, or as primitive in an iterative algorithm. A full discussion of these uses goes beyond the scope of this response.